# Multidimensional Predictors of Cancer-Related Fatigue Based on the Predisposing, Precipitating, and Perpetuating (3P) Model: A Systematic Review

**DOI:** 10.3390/cancers15245879

**Published:** 2023-12-17

**Authors:** Yiming Wang, Lv Tian, Xia Liu, Hao Zhang, Yongchun Tang, Hong Zhang, Wenbo Nie, Lisheng Wang

**Affiliations:** 1School of Nursing, Jilin University, No. 965 Xinjiang Street, Changchun 130021, China; wangyiming21@mails.jlu.edu.cn (Y.W.); tianlv21@mails.jlu.edu.cn (L.T.); 2Senior Department of Hematology, The Fifth Medical Center of PLA General Hospital, Beijing 100071, China; yoyoxia3588@sina.cn (X.L.); tangyc1309152820@163.com (Y.T.); zhanghong08021220@163.com (H.Z.); 3Yanda Medical Research Institute, Hebei Yanda Hospital, Sanhe 065201, China; 13691477201@163.com

**Keywords:** cancer-related fatigue, predictor, systematic review

## Abstract

**Simple Summary:**

Cancer-related fatigue (CRF) is the most common and distressing symptom in cancer survivors, severely affecting their quality of life. However, clinicians and patients are not well recognized for its importance and lack timely screening and assessment. With the rapid development of artificial intelligence and personalized care, early screening and assessment of CRF using machine learning to construct risk prediction models may contribute to this. Therefore, we redefined the predictors of CRF based on the predisposing, precipitating, and perpetuating (3P) model to develop a valid basis for the feature selection of future prediction models, intending to provide a more accurate and personalized plan for the clinical diagnosis and management of CRF.

**Abstract:**

Cancer-related fatigue (CRF) is a widespread symptom with high prevalence in cancer patients, seriously affecting their quality of life. In the context of precision care, constructing machine learning-based prediction models for early screening and assessment of CRF is beneficial to this situation. To further understand the predictors of CRF for model construction, we conducted a comprehensive search in PubMed, Web of Science, Embase, and Scopus databases, combining CRF with predictor-related terms. A total of 27 papers met the inclusion criteria. We evaluated the above studies into three subgroups following the predisposing, precipitating, and perpetuating (3P) factor model. (1) Predisposing factors—baseline fatigue, demographic characteristics, clinical characteristics, psychosocial traits and physical symptoms. (2) Precipitating factors—type and stage of chemotherapy, inflammatory factors, laboratory indicators and metabolic changes. (3) Perpetuating factors—a low level of physical activity and poorer nutritional status. Future research should prioritize large-scale prospective studies with emerging technologies to identify accurate predictors of CRF. The assessment and management of CRF should also focus on the above factors, especially the controllable precipitating factors, to improve the quality of life of cancer survivors.

## 1. Introduction

Cancer-related fatigue (CRF) is defined by the National Comprehensive Cancer Network (NCCN) as a distressing and persistent subjective feeling of exhaustion or tiredness that is cognitive, emotional, or physical. It is associated with cancer or cancer treatment that is not proportional to recent activity and interferes with usual functioning [1]. CRF is a frequently reported symptom, with a prevalence of 45 to 80% among all cancer patients [2,3], with a particular emphasis on those undergoing radiotherapy or chemotherapy [4].

The pathogenesis of CRF is unclear and may be related to inflammation, endocrine system disorders, adenosine triphosphate (ATP) metabolic abnormalities, 5-hydroxy tryptamine (5-HT) system dysfunction, and genetic factors [5,6,7,8,9,10,11]. The most popular and supported hypothesis includes the inflammatory hypothesis, which states that cancer and cancer treatments activate the immune system to release pro-inflammatory factors that affect the central nervous system, resulting in symptoms such as sleep disturbances, fever, and severe fatigue [5,7].

In the state of fatigue, the patient’s immune function is reduced and susceptible to infections, as well as feeling weak and discouraged, seriously affecting their therapeutic effect and quality of life and even increasing the risk of death [4,12,13]. Effective early screening and fatigue assessment are essential for these patients [14]. Although CRF management is strongly recommended in guidelines and the literature, its implementation in clinical practice is often lacking, leading to underestimation and undertreatment [15,16,17]. The primary barrier to implementing CRF management is the lack of accurate knowledge by care providers about fatigue and its treatment options and the effects of fatigue in patients [18]. Furthermore, patients often do not voluntarily report this symptom for fear of interfering with treatment or because they feel that fatigue is unavoidable [14,19].

In personalized treatment and care, precise and effective CRF prediction can impact the status quo and direct treatment decisions for patients and providers. With the rapid development of artificial intelligence, several studies have shown that prediction models based on machine learning algorithms can be a good aid for early screening and assessment of diseases [20] and have promising applications in CRF [21,22]. The selection of predictors, which serve the accuracy and interpretability of the model to a certain extent, is the primary foundation of constructing a prediction model. The selection of easily accessible electronic health records (EHRs) for modeling [23] purposes is a common practice observed in the previous research. Nevertheless, predictive models may need to be improved in their applicability and accuracy due to variations in biology, genetics, and environmental factors among different populations [24]. Thus, a comprehensive and systematic criterion for selecting predictors is required in similar studies in the future.

Sleight et al. proposed the predisposing, precipitating, and perpetuating (3P) factors model to facilitate risk prediction and clinical care of fatigue [25]. Predisposing factors are personal traits contributing to fatigue, such as biological behaviors like age, gender, and genetic variation, and also psychosocial factors like depression and anxiety. In this context, researchers have summarized the potentially significant associations of genetic polymorphisms associated with the neurotransmitter system, the hypothalamic–pituitary–adrenal (HPA) axis, and immune-mediated inflammation with fatigue [26]. Furthermore, Susanne et al. found that baseline levels of fatigue and depression were significant predictors of fatigue in all dimensions [27]. Precipitating factors stimulate the onset or change of fatigue, for instance, inflammatory changes caused by radiotherapy and chemotherapy. Raudonis et al. found that chemotherapy type and serum interleukin-6 (IL-6) were significant predictors of fatigue [8]. Perpetuating factors include poor sleep and chronic nutritional deficiencies, contributing to the exacerbation or gradual onset of fatigue. An increased risk of fatigue related to cancer was associated with a low recent protein intake, as determined via a 24 h recall conducted by Stobäus et al. [28].

In summary, there have been many attempts to predict cancer-related fatigue. However, they were single and one-sided, and studies still need to identify more comprehensive predictive markers of CRF based on the perspective of the occurrence and development of CRF. In this systematic review, we utilized the 3P model as a theoretical foundation to review various possible predictors of cancer-related fatigue other than the genomic domain to provide a broader and more personalized approach to the clinical diagnosis, treatment, and management of cancer-related fatigue.

## 2. Methods

### 2.1. Search Strategy

This study was conducted according to The Preferred Reporting Items for Systematic Reviews and Meta-Analyses 2020 (PRISMA 2020) guidelines [29]. The PubMed, Web of Science, Embase, and Scopus databases were thoroughly searched for relevant literature until 16 March 2023. The search strategy was explicitly adapted to the retrieval systems of the different databases, and the search was based on a combination of subject terms and accessible terms. Using search terms such as “cancer-related fatigue, predictor,” the search was limited to English articles and human participants, and the literature search was conducted independently by two researchers (Y.W. and L.T.), resulting in 27 papers. All search strategies were determined after several pre-searches (see Appendix A for details of the search strategies.) The PRISMA flowchart showed the detailed strategies for paper search and screening. Y.W. and L.T. independently evaluated the articles regarding the inclusion and exclusion criteria (see below), and any disagreement was discussed and negotiated to determine the final study for this review. This study has been registered on the Prospero website (CRD42023408601).

### 2.2. Inclusion and Exclusion Criteria

The inclusion criteria for this review were as follows:(1)Study participants were cancer patients with fatigue due to various cancer or cancer treatment (including solid and liquid tumors).(2)Studies which focused on the association of biomarker or risk factor with cancer-related fatigue.(3)Studies in which the primary outcome or secondary outcome was cancer-related fatigue.

The exclusion criteria were as follows:(1)Studies in which the outcome indicator was chronic fatigue syndrome or other disease-related fatigue rather than CRF.(2)Studies which did not report any correlation between biomarkers or risk factors and CRF.(3)Biomarker or risk factor of CRF was any gene polymorphism.(4)Studies that were not published in English or were not available in full text.(5)Reviews, meta-analyses, protocols, animal experiments, conference reports, medications, case reports, and non-human studies.(6)Duplication.

### 2.3. Data Extraction

In this study, data that satisfied the aforementioned criteria were extracted and saved in a Microsoft Excel spreadsheet by two researchers (Y.W. and X.L.). The primary outcome measures were predictors and assessments of CRF. The predictors were classified into predisposing, precipitating, and perpetuating factors according to the 3P model defined by Sleight et al. [25]. The precipitating and perpetuating factors were determined solely based on the above theory. According to the research of Hwang et al. [30], the predisposing factors were categorized into baseline fatigue, demographic characteristics, clinical characteristics, psychosocial traits, and physical symptoms. The following data were also extracted for each included study: author, year of publication, country of origin, study type, data source, sample size, cancer types, and definition of CRF.

### 2.4. Quality Assessment

The quality of cohort studies was assessed using the Newcastle–Ottawa Quality Assessment Scale (NOS) [31], including selection of study cohort populations, comparability between groups, and outcomes/exposures. Total scores ranged from 0 to 9, and poor-quality works were excluded (with a score ≥6 indicating high quality). The critical appraisal checklist from the Joanna Briggs Institute (JBI) was used to evaluate the quality of cross-sectional and longitudinal studies [32]. The questionnaire contains eight questions that were answered with yes, no, or unclear. A score of “yes” for >5 times, 3–4 times, and 0–2 times is considered high methodological quality, moderate methodological quality, and low methodological quality, respectively. The Cochrane Collaboration’s tool for assessing the risk of bias was used to evaluate the quality of randomized controlled trials (RCTs) [33], covering selection bias, performance bias, detection bias, attrition bias, reporting bias, and other biases. The assessment of study quality was independently conducted by two reviewers (Y.W. and H.Z.), and the results were compared until a consensus was reached. If a study received a low rating in all areas, it would be excluded from the review.

## 3. Results

### 3.1. Overview

An initial search of electronic databases identified 22,839 articles for review. After removing 6719 duplicates, 16,120 documents were retained. A total of 8892 articles were excluded based on title, keywords, and abstract, and 188 full-text articles were reviewed for eligibility. Ultimately, 27 articles published between 2002 and 2023 were included in this systematic review based on our inclusion criteria. The PRISMA flowchart is shown in Figure 1.

The studies were published between 2002 and 2023 (Figure 2B) and included eight cross-sectional studies, fifteen longitudinal studies, three cohort studies, and one RCT (Figure 2A). Studies were conducted primarily in the United States (*n* = 11), Canada (*n* = 4), China (*n* = 3), and Australia (*n* = 3) (Figure 2C). The number of patients included in the studies ranged from 11 to 3492 with various types of cancer, with breast cancer (*n* = 9) and mixed tumor populations (*n* = 8) accounting for approximately two-thirds of the total number of studies (Figure 2D). Data for the study were mainly collected using questionnaires (*n* = 26), electronic medical records (*n* = 22), and blood samples (*n* = 14) (Figure 2E). There was no standardized instrument for evaluating CRF, and most of the studies were assessed using reliability-tested scales, including the Functional Assessment of Cancer Therapy Fatigue (FACT-F) (*n* = 5), Multidimensional Fatigue Inventory-20 (MFI-20) (*n* = 5), Brief Fatigue Inventory (BFI) (*n* = 4), Functional Assessment of Chronic Illness Therapy-Fatigue (FACIT-F) (*n* = 2), Memorial Symptom Assessment Scale (MSAS) (*n* = 2), Piper Fatigue Scale Revised (PFS-R) (*n* = 2), Chalder Fatigue Questionnaire (CFQ) (*n* = 1), Cancer Fatigue Scale (CFS) (*n* = 1), European Organization for Research and Treatment of Cancer Quality of Life—fatigue assessment 12 item (EORTC QLQ-FA12) (*n* = 1), The Fatigue Symptom Inventory (FSI) (*n* = 1), Multidimensional Fatigue Symptom Inventory—Short Form (MFSI-SF) (*n* = 1), Somatic and Psychological Health Report questionnaire (SPHERE) (*n* = 1), and Verbal Numerical Rating (VNR) (*n* = 1) (Figure 2F).

Most of the studies analyzed fatigue scores as continuous variables. Based on this criterion, some of them transformed into categorical variables considering the minimal clinically important differences (MCIDs) as a cut-off score, differentiating between changes in fatigue as clinically significant or not, or using a boundary value to classify the severity of fatigue. A FACIT-Fatigue score less than 43 was considered fatigue [34], and a score less than 30 was severe fatigue [35]. A FACT-F standardized score of 68 or less was considered fatigue [36], and a change of at least 3 was deemed clinically significant [37,38,39]. A change in CFQ score greater than fourwas considered clinically significant [40]. In different studies, the designation of clinically significant fatigue (CSF) was defined as BFI scores equal to or greater than four [28], similar to VNR scores [41]. Hwang et al. used a threshold score of three to differentiate fatigue from non-fatigue [30].

Table 1 and Figure 2 depict the main characteristics of the included studies.

Y.W. and Y.T. evaluated the quality of the included studies separately, and found that most of them were of high quality. However, some cross-sectional and longitudinal studies failed to identify or did not address potential confounders; one cohort study had a pre-existing outcome event, fatigue, before the start of the study, and the only RCT did not mention whether it was blinded to the outcome of the study. Table 2, Table 3 and Table 4 demonstrate the quality assessment results of these studies.

### 3.2. Predisposing Factors

Predisposing factors in the identified studies focused on four areas: baseline fatigue, demographic characteristics, clinical characteristics, and psychosocial traits. Further details regarding these domains are provided below.

#### 3.2.1. Baseline Fatigue

Several studies have assessed the impact of baseline fatigue on cancer-related fatigue during the treatment cycle of patients. Baseline fatigue emerged as the strongest predictor in a longitudinal study involving 948 mixed cancer survivors with graded multiple linear regression analyses, using different dimensions of fatigue where total fatigue was considered as dependent variables and sociodemographic variables, clinical variables, psychological variables, and baseline fatigue as independent variables [27]. Patients with localized colorectal cancer who exhibited at baseline were more likely to have fatigue at follow-up investigations (6, 12, and 24 months after the start of chemotherapy) [36]. Patients who reported higher levels of fatigue after orchiectomy also reported higher levels of fatigue three months and one year after surgery [54]. In other words, early fatigue levels predicted their later levels.

#### 3.2.2. Demographic Characteristics

Sociodemographic characteristics of patients, such as biological sex and body composition, are crucial antecedents of cancer-related fatigue. Two studies showed that age correlates with CRF at various stages of cancer treatment, with higher-aged patients being more likely to develop fatigue than younger patients, implying that fatigue levels increase with age [50,54]. A study showed that female patients were twice as likely as males to develop moderate-to-severe fatigue using pathway analysis [50], implying that females are more susceptible to severe CRF. Results from another study indicated that a higher body mass index (BMI) ≥ 25 was a predictor of CSF in breast cancer patients after diagnosis and treatment [41].

#### 3.2.3. Clinical Characteristics

Clinical characteristics of patients, such as disease stage, tumor size, upper limb volume, and comorbidities, are also closely related to cancer-related fatigue. An independent predictor of fatigue was identified in a cross-sectional study of patients with hematologic neoplasms based on disease stage; patients who were newly diagnosed, refractory, or recurrent had a significantly higher likelihood of experiencing fatigue [46]. Results of a cohort study in breast cancer patients with up to 5 years of follow-up showed that among 48 women who developed persistent fatigue one month after treatment, a larger tumor size (>3 cm) was the only predictor of persistent CRF, whereas demographic variables, psychologic, surgical, or hematologic parameters were not valid predictors [49]. In another study, researchers showed that breast cancer patients with an increased upper limb volume (80% of total limb length measured from the ulnar styloid to the tip of the acromion) were associated with CSF as a predictor [41]. Furthermore, individuals with two or more comorbidities are nearly twice as likely as those with fewer comorbidities to have a severe form of CRF [50].

#### 3.2.4. Psychosocial Traits

The psychosocial traits associated with CRF in this review fall into three main categories: anxiety, depression, and other emotions. Two studies in 2019 concluded that depression impacts the development of fatigue, with significant differences in depressive symptom scores between fatigued and non-fatigued subjects, and predicts different dimensions of fatigue, such as cognitive, emotional, and physical [27,39]. Anxiety was the strongest predictor of mental fatigue and predicted the severity of fatigue in patients with testicular cancer one year after diagnosis and during chemotherapy for breast cancer [40,54].

Furthermore, several studies have found that patients’ psychological and behavioral responses can predict fatigue levels in distinct ways. Multivariate analyses in two studies showed that feeling sad and irritable predicted fatigue [30,46]. In breast cancer patients, increased fatigue was associated with various ineffective psycho-behavioral responses to symptoms and a decline in cognitive status, presented in three studies [36,40,56]. To be more precise, cancer-related catastrophes, all-or-nothing behaviors, avoidance behaviors, and perceived punishing responses from others are important variables in predicting fatigue [40], and stress process theory (cancer stressors and passive and active coping) might explain some of the variances in fatigue severity [56]. Similar results were obtained in two other studies of breast cancer patients, indicating that patients’ psychological patterns (mood disorders) significantly predicted CRF [53], and higher fatigue catastrophizing scores at baseline were also significant predictors of CRF prevalence [55].

#### 3.2.5. Physical Symptoms

The included literature identified pain, a lack of appetite, feeling drowsy, dyspnea, and urinary dysfunction as physical symptom dimensions that could predict fatigue. Hwang et al. explored the sources of predictor variables of fatigue in different dimensions. They found that physical symptoms such as pain, a lack of appetite, feeling drowsy, and dyspnea served as a unidimensional predictor of fatigue in patients with mixed cancers and functioned as the sole independent predictor in the multidimensional model [30]. The results of another three studies in patients with different types of cancer showed that pain is positively correlated with fatigue severity [29,37,40]; however, pain cannot be used as an independent predictor of fatigue.

### 3.3. Precipitating Factors

For precipitating factors, most of the included studies focused on exploring the underlying immune and inflammatory factors for CRF, with the correlation between proinflammatory factors and fatigue severity being the most commonly examined, apart from the effect of radiotherapy and chemotherapy on CRF, which produced mixed results.

#### 3.3.1. Radiotherapy and Chemotherapy

Fatigue was investigated in a subset of the literature in the analysis of treatment procedures, including chemotherapy and radiotherapy. Several studies have documented patients who are undergoing chemotherapy for nasopharyngeal cancer [44], breast cancer [8,55], colorectal cancer [36], and head and neck cancer [45] experiencing increased fatigue to differing degrees. Ahlberg et al. conducted a similar study on 15 patients with pelvic cancer undergoing radiotherapy, they found that fatigue increased as the radiotherapy progressed [57]. Conversely, another study found no statistically significant distinction in the prevalence of fatigue between the chemotherapy and chemotherapy/radiotherapy groups, implying that the high incidence of fatigue cannot be attributed to radiotherapy alone but rather to chemotherapy [47].

Fatigue is influenced to different extents by both the type and course of chemotherapy. According to a study [8], patients with breast cancer who underwent chemotherapy for an extended period of time experienced the highest degree of fatigue before the end of chemotherapy. Specifically, patients on Adriamycin/Cytoxan dose-intensive and Taxol dose-intensive chemotherapy regimens had a lower likelihood of experiencing fatigue than those on Taxotere, Cytoxan, and Adriamycin chemotherapy regimens. Another study found that the chemotherapeutic agent was the only variable significantly affecting different fatigue levels. Patients who received cisplatin experienced notably reduced fatigue levels one month after intensity-modulated radiotherapy (IMRT) compared to those who received carboplatin/paclitaxel [45]. Another study concluded that patients receiving 3–6 courses of treatment were at a higher risk of severe fatigue than other patients and that this factor predicted severe overall fatigue [44].

#### 3.3.2. Inflammatory Factors

The CRF predictors categorized as precipitating factors were concentrated on inflammation and immune response. In the present studies, the following factors were primarily evaluated: (1) proinflammatory cytokines IL-1 [47,53], IL-6 [8,57], tumor necrosis factor-α (TNF-α) [47], IL-8, IL-16, Interferon-α (IFN-α2), IFN-γ; (2) anti-inflammatory cytokines IL-10; (3) hematopoietic cytokines IL-3, IL-9; (4) chemokines: interferon gamma-inducible protein 10 (IP-10), stromal cell-derived factor 1α (SDF-1α) [37].

Two studies found that elevated plasma levels of IL-1 were significantly associated with CRF in tumor patients and IL-1β was a predictor of cancer-related fatigue during adjuvant chemotherapy [47,53]. Another proinflammatory factor, TNF-a, was also a predictor of CRF [47]. However, Ahlberg et al. found different results when monitoring levels of fatigue and inflammatory factors before and during radiotherapy in patients with pelvic cancer: fatigue increased as radiotherapy progressed, but the changes in IL-1 and TNF were insignificant. They did not correlate with the degree of fatigue, and found a significant negative correlation between IL6 and the general fatigue dimension [57]. Raudonis et al. obtained the same conclusion regarding breast cancer patients and concluded that IL-6 was a significant predictor of fatigue [8].

A subset of studies has investigated the potential predictive value of alternative cytokine types other than pro-inflammatory factors on CSF. Feng et al. found that during extracorporeal radiation therapy (EBRT) in patients with prostate cancer, in addition to proinflammatory factors IL-8, IL-16, IFN-α2, and IFN-γ, IL-10, IL-3, IL-9, IP-10, and SDF-1α were also positively correlated with changes in fatigue scores from T1 (before the start of EBRT) to T3 (one year after EBRT), which can be used as early biomarkers to predict chronic fatigue [37].

#### 3.3.3. Laboratory Indicators and Metabolites

Chemotherapeutic drugs influence laboratory examinations such as routine blood tests and serum metabolites during the treatment of tumor patients and have received more attention in exploring predictors of CRF.

One of the contributors to the development of CRF is anemia, which accompanies the disease and its treatment. Cella et al. examined the differences in fatigue levels between cancer patients (anemic and non-anemic) and the general population in a large cohort study, using hemoglobin as the dependent variable in a multivariate regression analysis, showing that hemoglobin was the only significant predictor [58]. Another study conducted in a tertiary care hospital showed that among palliative care patients with advanced cancer, albumin levels were one of the main predictors of fatigue, whereas hemoglobin levels were not [35].

C-reactive protein (CRP), a well-recognized marker of systemic inflammation, is released in the early stages of inflammation. Two studies found that more severe fatigue is strongly associated with higher concentrations of serum CRP, which can serve as an independent predictor of fatigue severity in patients with breast cancer and multiple myeloma [48,52]. Two additional studies involving populations of breast cancer patients found that a white blood cell count increase beyond the baseline, exceeding 8000, was associated with persistent CSF [41]. Moreover, morning cortisol levels significantly predicted CRF during and after adjuvant therapy in patients [53]. In patients with multiple myeloma, a substantial correlation was observed between the serum metabolite guanidine acetic acid (GAA) and patient fatigue, with a statistically significant difference [42]. Moreover, another study explored the longitudinal link between glucocorticoids, a key regulator of the anti-inflammatory response, regarding their receptor sensitivity and fatigue. It was found that glucocorticoid receptor (GR) sensitivity throughout the sample during IMRT increased. Patients who experienced a gradual reduction in fatigue exhibited the greatest improvement in GR sensitivity over time. A lesser increase in GR sensitivity significantly predicted an increase in fatigue [45].

### 3.4. Perpetuating Factors

Poor exercise and dietary habits are persistent factors that influence cancer-related fatigue in patients. Fatigue has been linked to the physical activity levels of patients in three investigations [35,41,51]. Specifically, patients with higher levels of fatigue are less physically active, which may subsequently lead to body dysmorphic disorder, exacerbating the persistence of fatigue [59]. Fatigue at three and six months within the age group of elderly patients can be predicted by their baseline physical activity level [51]. Two studies have shown that survivors with a better nutritional status have relatively lower levels of fatigue [28,43]. Patients with CRF had significantly lower protein and energy intake than those without CRF. Moreover, recent inadequate protein intake (<1 g/kg body weight) and protein intake (g/kg body weight) were significant contributors to cancer-related fatigue [28].

## 4. Discussion

This systematic review aimed to identify predictors of cancer-related fatigue and categorize them based on the complex biological and psychological processes behind CRF using the 3P model as a framework (Figure 3). We found that predisposing factors (baseline fatigue, demographic characteristics, clinical characteristics, psychosocial traits, physical symptoms), precipitating factors (radiotherapy and chemotherapy, inflammatory factors, laboratory indicators, and metabolites), and perpetuating factors (physical activity levels, nutritional status) predicted the development of CRF to varying degrees.

### 4.1. Predisposing Factors

In this review, predisposing factors primarily pertain to the following factors: (1) baseline fatigue; (2) demographic characteristics such as older age, BMI ≥ 25, and female gender; (3) clinical characteristics such as stage of disease, larger tumor volume, increased upper limb volume, and multiple comorbidities; (4) psychosocial traits such as anxiety, depression, pain, and negative psychological and behavioral responses; (5) physical symptoms including pain, lack of appetite, feeling drowsy, dyspnea, and urinary dysfunction.

We found that the demographic characteristics of high age, BMI ≥25, and female are predictors of CRF in patients, consistent with previous findings [50,54,60,61]. However, one study found lower BMI to be a predictor of general fatigue [44], implying that low BMI represents the poorer nutritional status of the patient. Furthermore, one study found that in the breast cancer population, younger rather than older patients reported more severe fatigue and poor quality of life [62], which may be related to their experience of sudden menopause and more severe psychological fallout [63]. Moreover, literacy level and occupation are also associated with fatigue. In this context, Fleer et al. found that testicular cancer patients with higher levels of education reported more symptoms on the dimensions of general and physical fatigue [54]. Chen et al. found that unemployed participants experienced more mental fatigue than those engaged in occupations [44]; however, it could not be used as a significant predictor of CRF. Knowing that patients with such characteristics will experience severe CRF is beneficial because it allows clinicians to focus on such patient populations in advance and provide preventive measures.

Similarly, disease characteristics with a worse prognosis also imply more severe fatigue. Patients with refractory or recurrent disease and those with more extensive tumor volumes are significantly more likely to experience fatigue [35,46], which may be related to increased tumor burden, compromised body systems, and tumor-associated inflammatory responses. Increased upper extremity volume in breast cancer patients predicts CSF [41], which may also be associated with an inflammatory response, where the edema exudate may contain high levels of proinflammatory cytokines [64]. Furthermore, patients with multiple baseline comorbidities have more severe fatigue [50,51], regardless of age. More severe disease conditions increase the physical burden on patients and imply more severe fatigue.

Apart from fatigue, cancer patients frequently experience various long-standing symptoms, such as pain, depression, and anxiety. These symptoms are correlated and can interact [65,66] and synergistically negatively impact the quality of life for cancer survivors [67]. The four studies in this review showed that pain positively correlated with fatigue severity [30,35,39,56]. Moreover, previous clinical investigations have observed that pain and depression are frequently comorbid [68] and often respond to the same treatments and exacerbate each other [69]. Thus, it is not difficult to understand that depression can predict CRF [27,39], which is consistent with the findings of Vardy et al. [36] and Andrykowski et al. [55], whereby increased depressive symptoms were found to correspond to more fatigue. However, using multiple logistic regression, Hwang et al. determined independent predictors of CSF in 180 oncology patients; they found that the Zung Self-Depression Scale (SDS) score, which represents depression, was not an independent predictor [30]. Moreover, anxiety [40,54], feeling sad and feeling irritable [30,46], fatigue catastrophizing [55], stress passive coping [56], mood disorders [53], cancer-related catastrophes, all-or-nothing behaviors, avoidance behaviors, and perceived punishing responses from others [40] can be significant variables in predicting fatigue. Most of the studies mentioned above were conducted on breast cancer patient populations [36,40,53,55,56]. A recent review reaffirmed that CRF is the most prevalent and distressing symptom in breast cancer survivors after treatment [70]. Moreover, another study highlighted that breast cancer-related fatigue (BCRF) is a multidimensional concept that impacts physical, cognitive, and emotional domains [71]. There is growing evidence that the perception and experience of BCRF are related to multiple psychological factors [70,72]; however, a more comprehensive overview has not been established in this area, which necessitates further research.

In summary, patients with these traits are more likely to have fatigue. For all categories of people, baseline fatigue often predicts continued feelings of fatigue in subsequent periods [27,36,54]. A study found that self-reported fatigue represents a poor prognosis for older patients with hematologic malignancies [73], implying that patients with higher levels of fatigue also have a more significant overall symptom burden. Identifying fatigued individuals early in the disease trajectory is required to reduce post-treatment fatigue, which has significant implications for our understanding of fatigue development and management.

### 4.2. Precipitating Factors

Precipitating factors are states and features that cause or accelerate the onset of cancer-related fatigue. In this review, the following are classified as predictors of CRF precipitating factors: (1) radiotherapy and chemotherapy and related symptoms, (2) inflammatory factors, (3) laboratory indicators and metabolites.

As shown in the introduction, the mechanism of CRF needs to be clarified, related to finely accelerated aging [74,75] (e.g., premature shortening of telomeres and alterations in DNA methylation), inflammation (overproduction of proinflammatory cytokines), and metabolic dysregulation (alterations in metabolic genes and regulatory pathways) due to radiotherapy [25]. Fatigue is exacerbated in patients during radiotherapy and chemotherapy [8,36,44,45,55,57], and the type [8,45] and course [44] of chemotherapy also affect fatigue to varying degrees. Inflammation may be responsible for this phenomenon as a critical biological pathway leading to CRF [76,77]. Fatigue is strongly associated with proinflammatory cytokines, including IL-1 [47,53], IL-6 [8,57], and TNF-α [47], according to several review studies. The result is consistent with a recent finding that biological pathways such as polymorphisms in inflammatory risk genes, alterations in the HPA axis, and alterations in the cellular immune system regulate the production of proinflammatory factors [77]. Thus, many cytokine receptors are located in the hypothalamus, which is richly connected to the brainstem, frontal cortex, and limbic system. These brain regions further influence mood, behavior, and motor flexibility. In essence, the neuroimmune interaction leads to the disease behavior represented by CRF [78].

Furthermore, immune system-related markers such as white blood cells (WBCs) [41], hemoglobin [58], albumin [35], CRP [48,52], and cortisol [53] can be considered independent predictors of CRF. Although lymphocyte count does not predict overall fatigue, it is significantly related to general and mental fatigue and a risk indicator for predicting mental fatigue [44], consistent with previous findings which found that altered lymphocyte subsets and anemia are associated with fatigue during treatment [76,79,80]. As part of the immune system, WBCs usually increase in number when inflammation or infection occurs, and simultaneously, CRP produced by the liver is released into the bloodstream. Maurer et al. suggested that higher levels of CRP are often associated with CRF [81], consistent with the outcomes of the present review. Hemoglobin levels can directly determine the oxygen-carrying capacity of red blood cells, where low levels can lead to anemia and insufficient oxygen supply, fatigue, and is strongly associated with fatigue [80]. Conversely, albumin level is closely related to the nutritional status of the patient, since protein intake is disrupted by cancer and radiotherapy-induced nausea, vomiting, and loss of appetite, which consequently lead to fatigue [82,83]; studies showed that improved nutritional status can change this situation [84]. As for cortisol-predicting CRF in breast cancer patients [53], the HPA axis system controls cortisol release [85], and its secretion usually follows a diurnal pattern. Previous studies suggested that lower cortisol in the morning indicates HPA dysregulation [86], affecting the activity of the immune system and manifests itself in patients as increased fatigue [87].

Furthermore, the questionnaires and clinical data which comprised the majority of information sources for the studies included in this review, as well as a subset of the patient’s blood samples, were collected to examine for conventional markers such as inflammatory factors. However, one study used emerging metabolomic techniques to analyze serum metabolites in patients with multiple myeloma and found that GAA could predict fatigue [42]. Metabolomics which follows in the footsteps of the “big three” (genomics, transcriptomics, and proteomics) is an emerging histology and is presently utilized extensively in disease diagnosis and personalized therapy [88]. Studies have shown that the magnitude of fatigue is related to metabolic patterns [89], and metabolomics-based CRF analysis can help to elucidate the mechanisms underlying CRF [90].

### 4.3. Perpetuating Factors

Perpetuating factors are characteristics and behaviors that exacerbate or prolong CRF. Low physical activity levels [35,41,51] and poor nutritional status [28,43] were categorized as the factors in the studies included in this review.

Gerber et al. found that low levels of physical activity in patients with primary breast cancer were correlated with persistent CSF [41], consistent with the findings of another study in this area of 440 older mixed cancer patients [51] which demonstrated that the poorer physical functioning and performance status of the patients was a significant predictor of CRF [35]. Factors related to cancer and treatment may initially contribute to acute fatigue and limit the patient’s daily activities to some extent, and then reduced physical activity may lead to a decline in physical fitness, contributing to the long-term persistence of CRF [35,78]. Further longitudinal studies are required to assess the temporal associations between fatigue, physical inactivity, and susceptibility to fatigue in cancer survivors and examine the existence of a potential causal relationship [91]. Moreover, the nutritional status of the patient has also shown to be associated with fatigue [28,43], consistent with the results of one of the largest fish oil supplementation trials conducted among breast cancer survivors, where supplementation showed more significant improvements in the physical fatigue and vigor dimensions of MFSI. Furthermore, survivors with better nutritional status had a tremendous increase in total serum omega-3 fatty acids from fish oil supplementation, which positively moderated its effects on cancer-related fatigue [43]. A study of colorectal cancer patients also found that laboratory markers of nutritional status were strongly associated with CRF [92]. Existing research indicates that fatigue in cancer populations can be moderated by nutritional status. Furthermore, dietary interventions and improved nutritional status have been linked to numerous health benefits [93]. Therefore, addressing malnutrition and conducting screenings could be an excellent starting point for reducing patient fatigue.

### 4.4. Limitations and Prospects

This is the first systematic review to categorize cancer-related fatigue predictors in terms of the 3P model, which will facilitate future researchers to comprehend and characterize CRF from the perspective of complex biological and psychological mechanisms behind CRF. However, we could not draw firm conclusions based on the present research due to the prevailing gaps in our knowledge of fatigue and the limitations of the included studies.

We found that the outcomes of the exact predictor were only somewhat consistent across studies; this may be attributable to the heterogeneity of the study populations and the different measurement instruments used. In all the studies included in this systematic review, cancer-related fatigue was measured by utilizing 13 different instruments, with the most commonly used instruments being the FACT-F, MFI-20, and BFI. Unidimensional questionnaires similar to the BFI have fewer questions, which are easier to use. However, the lack of predictors for different dimensions of fatigue has somewhat hindered the exploration of predictors for different dimensions of fatigue using the 3P model; out of the 27 studies included, only 11 defined fatigue, which needed to be more consistent. Most definitions differentiate the presence or absence of fatigue by a specific cut-off value of the questionnaire scores. Agarwal et al. and Raudonis et al. differentiated the severity of fatigue [8,35]. In contrast, Feng et al. compared the value of change in fatigue using the minimum clinically important difference to examine a clinically meaningful difference [37,39]. The lack of uniform definitional criteria to define fatigue may lead to non-comparability and confounding of results. A recent meta-analysis showed that the incidence of cancer-related fatigue measured using the different study instruments varied, which may be associated with inconsistencies in the sensitivity and specificity of the scales [94]. However, the diagnosis and assessment of cancer-related fatigue is currently inconsistent globally, and it is recommended that clinical practitioners establish a benchmark for this evaluation to increase its comparability and dependability.

Furthermore, most of the information obtained from the literature was found in cross-sectional and longitudinal studies without a control group; thus, we could not determine whether the associations between these predictors and fatigue were specific to disease diagnosis and/or treatment. Fatigue is a common symptom in the community, with up to half of the general population reporting fatigue in extensive surveys rather than being specific to cancer patients. Chronic fatigue syndrome (CFS) is a medical condition distinguished by severe and incapacitating fatigue that persists for a minimum of six months and is accompanied by various symptoms like rheumatologic, infectious, and neuropsychiatric [95]. While there may be similarities with the mechanisms underlying CRF, such as inflammation and permanent deconditioning [96], they are not precise. Hence, it could be advantageous to conduct a comparative analysis of CRF results and fatigue in other prolonged conditions using the 3P model as a framework to determine whether significant distinctions exist.

Finally, considering the above limitations, this review has three other areas of improvement. First, our search was limited to four extensively utilized databases, potentially leaving out ongoing unpublished studies and studies from other databases. Second, we should have included works studying the association of genetic polymorphisms with CRF because research involving these aspects already exists [26]; however, we incorporated comprehensive demographic, physical, and psychological indicators to the greatest extent possible. Third, because the approaches and definitions of assessing fatigue were not entirely consistent across studies, we performed only descriptive summaries and did not perform any statistical analyses.

## 5. Conclusions

This systematic review summarized for the first time the independent predictors of cancer-related fatigue based on the theoretical framework of the 3P model. (1) Predisposing factors—baseline fatigue; demographic characteristics such as older age, BMI ≥ 25, and females; clinical characteristics such as poorer disease stage, larger tumor volume, multiple comorbidities; psychosocial traits such as anxiety, depression, and physical symptoms such as pain and urinary dysfunction. (2) Precipitating factors—type and stage of chemotherapy, IL-1, IL-6, TNF-a, IL-8, IL-16, IFN-α2, IFN-γ, IL-10, IL-3, IL -9, IP-10, and SDF-1α and other inflammatory factors; laboratory markers such as WBC, hemoglobin, albumin, and CRP; metabolic changes such as GAA, cortisol levels, and GR sensitivity. (3) Perpetuating factors—a low level of physical activity and poorer nutritional status. These findings suggest that future management of cancer-related fatigue should focus on the above risk factors, especially the controllable precipitating factors.

Future research is anticipated to incorporate additional emerging technologies, such as genomics and metabolomics, to identify accurate CRF predictors grounded in a unified definition of CRF. This will be complemented by a greater number of large-scale prospective studies that investigate and validate these predictors, thereby establishing a theoretical foundation for clinical practice involving comprehensive and personalized management and treatment of CRF.

## Figures and Tables

**Figure 1 cancers-15-05879-f001:**
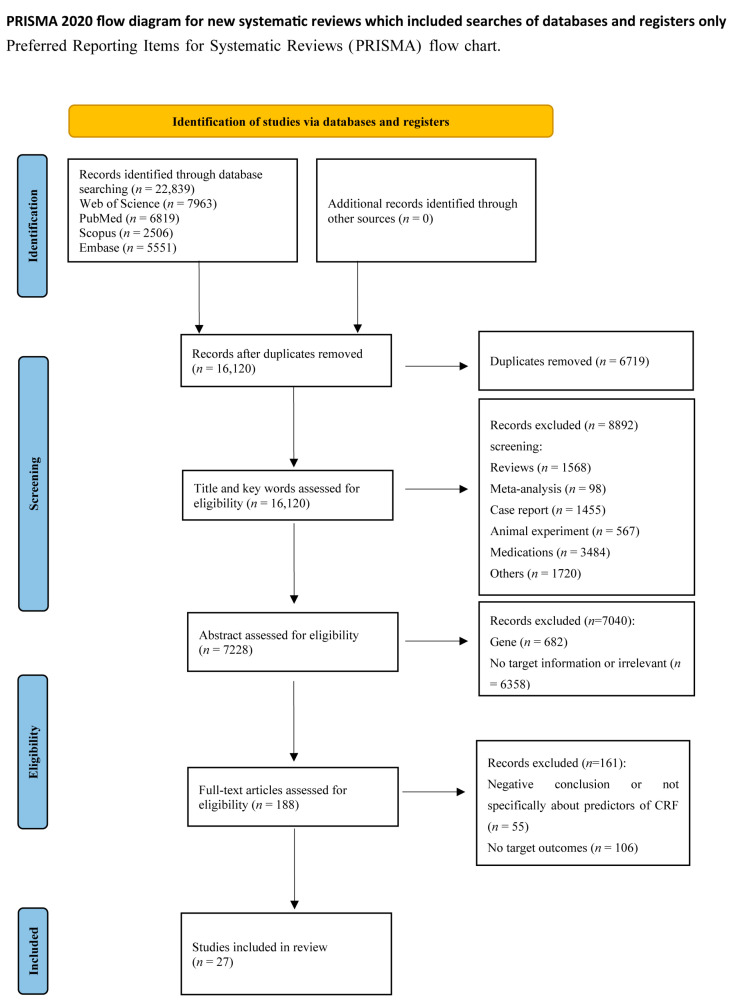
PRISMA diagram for study selection.

**Figure 2 cancers-15-05879-f002:**
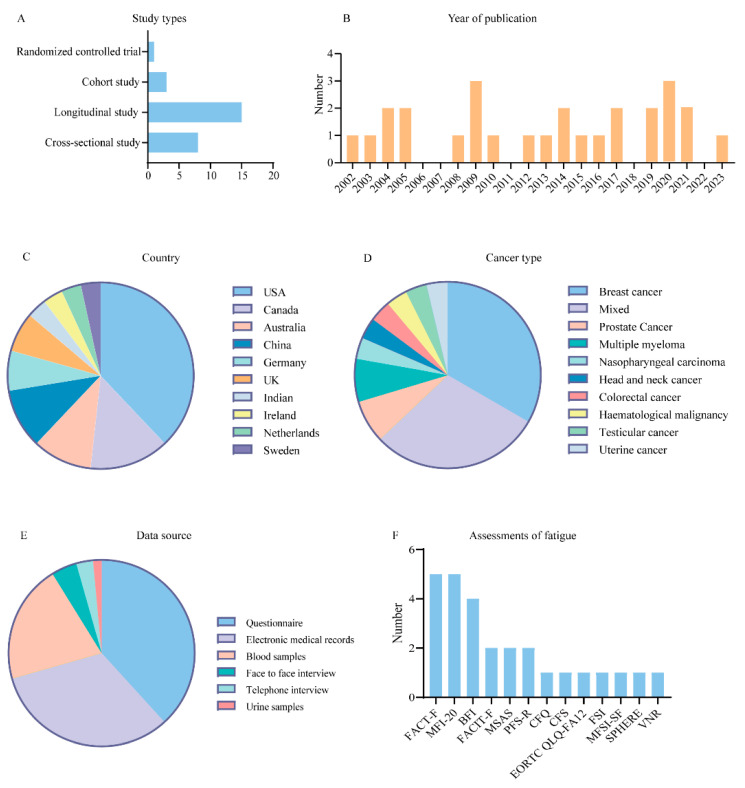
Characteristics of the included studies. (**A**) Bar chart for types of study of included studies. (**B**) Bar chart for numbers of publication years of included studies. (**C**) Pie chart for countries of included studies. (**D**) Pie chart for types of cancer of included studies. (**E**) Pie chart for data source of included studies. (**F**) Bar chart for fatigue assessment scales of included studies.

**Figure 3 cancers-15-05879-f003:**
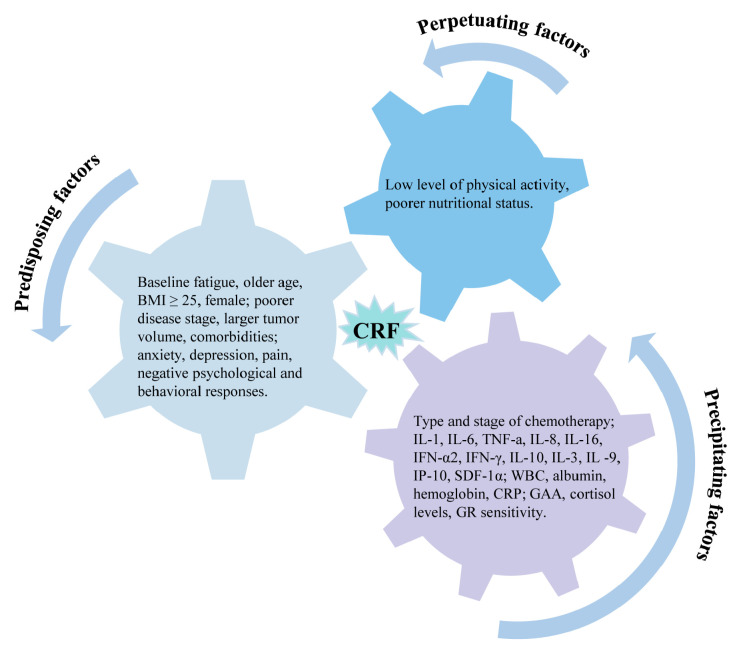
Predictors of cancer-related fatigue based on 3P factor model.

**Table 1 cancers-15-05879-t001:** Table of study characteristics, predictors, and assessments of cancer-related fatigue.

No.	Author, Year	Country	Study Type	Data Source	Sample Size	Cancer	Predictors	Assessments of Fatigue	Definition of Fatigue
1	Zhang, 2023 [42]	China	A cross-sectional study	A, B, C	30	Multiple myeloma	GAA	BFI-C	N/A
2	Kleckner, 2021 [43]	USA	RCT	A, B, C	85	Breast cancer	Serum omega-3s	MFSI-SF	N/A
3	Chen, 2021 [44]	China	A longitudinal study	A, B	79	Nasopharyngeal carcinoma	Cancer stage IVB, 3–6 courses of treatment	MFI-20	N/A
4	Xiao, 2020 [45]	USA	A longitudinal study	A, B, C	77	Head and neck cancer	GR sensitivity	MFI-20	N/A
5	Agarwal, 2020 [35]	India	A cross-sectional study	A	110	Mixed	Pain, physical functioning, performance status, albumin	FACIT-F	A score < 30 indicates severe fatigue
6	Hughes, 2020 [40]	UK	A longitudinal study	A, B	159	Breast cancer	Cancer-related catastrophizing, all-or-nothing behaviors, perceived punishing responses, anxiety	CFQ	A score > 4
7	Susanne, 2019 [27]	Germany	A longitudinal study	A, B	948	Mixed	Baseline fatigue, depression	EORTC QLQ-FA12	N/A
8	Feng, 2019 [39]	USA	A longitudinal study	A, B	47	Prostate Cancer	Urinary dysfunction, pain, depressive symptoms	FACT-F	A clinically significant decrease in the FACT-F score of ≥3 points
9	Raudonis, 2017 [8]	USA	A longitudinal study	A, B, C	11	Breast cancer	Chemotherapy type, time (sequence of visit), IL-6	PFS-R	Levels of fatigue range from 0 (absent), 0.1 to 3.99 (mild), 4 to 6.99 (moderate), or 7.0 or greater (severe)
10	Feng, 2017 [37]	USA	A longitudinal study	A, B, C	34	Prostate Cancer	IL-3, IL-8, IL-9, IL-10, IL-16, IP-10, IFN-α2, IFN-γ, SDF-1α	FACT-F	A score change of ≥3 is considered clinically significant
11	Vardy, 2016 [36]	Australia, Canada	A cohort study	A, B, C	361	Colorectal cancer	Baseline fatigue, cognitive and affective symptoms, quality of life, comorbidities, chemotherapy	FACT-F	Standardized score ≤ 68/100
12	Stobäus, 2015 [28]	Germany	A cross-sectional study	A, B, C, D	285	Mixed	Low recent protein intake	BFI	A score ≥ 4
13	Zordan, 2014 [46]	Australia	A cross-sectional study	A, B	180	Hematological malignancy	Performance status, stage of disease, feeling sad, feeling irritable	MSAS-SF	N/A
14	Zhang, 2014 [47]	China	A longitudinal study	A, C, E	200	Mixed	TNF-a, IL-1	CFS	A score ≥ 5
15	Pertl, 2013 [48]	Ireland	A longitudinal study	A, B, C	61	Breast cancer	CRP	FACT-F	N/A
16	Goldstein, 2012 [49]	Australia	A cohort study	A, B, C, D	218	Breast cancer	Tumor size	Fatigue subscale of SPHERE	A score ≥ 3
17	Gerber, 2011 [41]	USA	A longitudinal study	A, B, C	44	Breast cancer	BMI, WBC, upper limb volume, physical activity levels	VNR	A score ≥ 4
18	Hoffman, 2009 [50]	USA	A cross-sectional study	A, B	298	Mixed	Older age, comorbidity, female	BFI	N/A
19	Luctkar-Flude, 2009 [51]	Canada	A longitudinal study	A, B	440	Mixed	Physical activity levels	MSAS	N/A
20	Booker, 2009 [52]	Canada	A cross-sectional study	A, B	56	Multiple myeloma	CRP	FACT-F	N/A
21	Von Ah, 2008 [53]	USA	A longitudinal study	A, C	44	Breast cancer	Morning cortisol, IL-1β, mood disturbance	PFS-R	N/A
22	Fleer, 2005 [54]	Netherlands	A longitudinal study	A, B, C	52	Testicular cancer	Older age, trait anxiety, baseline fatigue	MFI-20	N/A
23	Andrykowski, 2005 [55]	USA, UK	A longitudinal study	A, B, D, E	288	Breast cancer	Chemotherapy, fatigue catastrophizing	FSI	N/A
24	Gélinas, 2004 [56]	Canada	A cross-sectional study	A, B	103	Breast cancer	Cancer-related stressors, passive and active coping, pain	MFI-20	N/A
25	Ahlberg, 2004 [57]	Sweden	A longitudinal study	A, B, C	15	Uterine cancer	IL-6	MFI-20	N/A
26	Hwang, 2003 [30]	USA	A cross-sectional study	A	180	Mixed	Physical symptoms (pain, lack of appetite, feeling drowsy, dyspnea); psychological symptoms (feeling sad and feeling irritable)	BFI, FACT-F	BFI usual fatigue ≥ 3/10
27	Cella, 2002 [58]	USA	A cohort study	E	3492	Mixed	Hemoglobin	FACIT-F	N/A

A = Questionnaire; B = electronic medical records; C = blood samples; D = face-to-face interview; E = telephone interview; F = urine samples. RCT, randomized controlled trial; BMI, body mass index; CRP, C-reactive protein; IFN, interferon; IL, interleukin; IP-10, interferon gamma-inducible protein-10; GAA, guanidine acetic acid; GR, glucocorticoid receptor; SDF, stromal cell-derived factor; TNF, tumor necrosis factor; WBC, white blood cell; BFI, Brief Fatigue Inventory; BFI-C, Chinese version of the Brief Fatigue Inventory; CFQ, Chalder Fatigue Questionnaire; CFS, Cancer Fatigue Scale; EORTC QLQ-FA12, European Organization for Research and Treatment of Cancer Quality of Life—fatigue assessment 12 item; FACIT-F, Functional Assessment of Chronic Illness Therapy-Fatigue; FACT-F, Functional Assessment of Cancer Therapy Fatigue; FSI, The Fatigue Symptom Inventory; MFI-20, Multidimensional Fatigue Inventory-20; MFSI-SF, Multidimensional Fatigue Symptom Inventory—Short Form; MSAS, Memorial Symptom Assessment Scale; MSAS-SF, Memorial Symptom Assessment Scale—Short Form; PFS-R, Piper Fatigue Scale Revised; SPHERE, Somatic and Psychological Health Report questionnaire; VNR, Verbal Numerical Rating; N/A, not available.

**Table 2 cancers-15-05879-t002:** Joanna Briggs Institute (JBI) scale of 23 studies in the systematic review.

Author, Year	A	B	C	D	E	F	G	H
Zhang, 2023 [42]	yes	yes	yes	yes	yes	no	yes	yes
Chen, 2021 [44]	yes	yes	yes	yes	yes	yes	yes	yes
Xiao, 2020 [45]	yes	yes	yes	yes	yes	yes	yes	yes
Agarwal, 2020 [35]	yes	yes	yes	yes	yes	yes	yes	yes
Hughes, 2020 [40]	yes	yes	yes	yes	yes	yes	yes	yes
Susanne, 2019 [27]	yes	yes	yes	yes	yes	yes	yes	yes
Feng, 2019 [39]	yes	yes	yes	yes	yes	yes	yes	yes
Raudonis, 2017 [8]	yes	yes	yes	yes	yes	yes	yes	yes
Feng, 2017 [37]	yes	yes	yes	yes	yes	no	yes	yes
Stobäus, 2015 [28]	yes	yes	yes	yes	yes	yes	yes	yes
Zordan, 2014 [46]	yes	yes	yes	yes	yes	yes	yes	yes
Zhang, 2014 [47]	yes	yes	yes	yes	yes	no	yes	yes
Pertl, 2013 [48]	yes	yes	yes	yes	yes	yes	yes	yes
Gerber, 2011 [41]	yes	yes	yes	yes	yes	no	yes	yes
Hoffman, 2009 [50]	yes	yes	yes	yes	yes	yes	yes	yes
Luctkar-Flude, 2009 [51]	yes	yes	yes	yes	yes	yes	yes	yes
Booker, 2009 [52]	yes	yes	yes	yes	yes	yes	yes	yes
Von Ah, 2008 [53]	yes	yes	yes	yes	yes	yes	yes	yes
Fleer, 2005 [54]	yes	yes	yes	yes	yes	no	yes	yes
Andrykowski, 2005 [55]	yes	yes	yes	yes	yes	yes	yes	yes
Gélinas, 2004 [56]	yes	yes	yes	yes	yes	yes	yes	yes
Ahlberg, 2004 [57]	yes	yes	yes	yes	no	no	yes	yes
Hwang, 2003 [30]	yes	yes	yes	yes	yes	yes	yes	yes

A. Were the criteria for inclusion in the sample clearly defined? B. Were the study subjects and the setting described in detail? C. Was the exposure measured in a valid and reliable way? D. Were objective, standard criteria used for measurement of the condition? E. Were confounding factors identified? F. Were strategies to deal with confounding factors stated? G. Were the outcomes measured in a valid and reliable way? H. Was appropriate statistical analysis used? Answers: yes, no, unclear, or not applicable.

**Table 3 cancers-15-05879-t003:** Newcastle–Ottawa Scale of 3 studies in the systematic review.

Author, Year	Selection				Comparability	Outcome			Total
	Exposed Cohort	Non-Exposed Cohort	Ascertainment of Exposure	Outcome of Interest		Assessment of Outcome	Length of Follow-Up	Adequacy of Follow-Up	
Vardy, 2016 [36]	1	1	1	0	2	1	1	1	8
Goldstein, 2012 [49]	1	1	1	1	2	1	1	1	9
Cella, 2002 [58]	1	1	1	0	2	1	1	1	8

**Table 4 cancers-15-05879-t004:** Cochrane Collaboration’s tool of one study in the systematic review.

Author, Year	Selection Bias		Performance Bias	Detection Bias	Attrition Bias	Reporting Bias	Other Bias
	Random Sequence Generation	Allocation Concealment					
Kleckner, 2021 [43]	low	low	low	unclear	low	low	low

Answers: low risk of bias, high risk of bias, unclear risk of bias.

## Data Availability

The data presented in this study are available in this article.

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
