# Peer review of "Multidimensional Predictors of Cancer-Related Fatigue Based on the Predisposing, Precipitating, and Perpetuating (3P) Model: A Systematic Review"

_cancers, 2023, doi:10.3390/cancers15245879_

Round 1

Reviewer 1 Report

Comments and Suggestions for Authors

A well presented manuscript representing a systematic review on identifying and categorization of predictors of CRF based on a theoretical framework of 3P model i.e. predisposing factors, precipitating factors and perpetuating factors.

It is correct, that the results of this review could stimulate future research employing emerging technologies to provide more refined theoretical basis of assessment and screening of CRF and this in turn could lead to a more accurate and personalized treatment of clinical work.

Despite the limitations elaborated in the paper, particularly the need for additional studies to corroborate the value of defined CRF predictors, it is plausible, as suggested, that assessment and management of CRF should also focus on factors of the 3P model especially the controllable predisposing factors to improve the quality of life of cancer survivors.

The approach is scientifically sound and research methods are appropriate. However, though the results are well presented there are areas where interpretations and conclusions are included in the same results instead of being in the discussion and or conclusion. On the other hand, the figures and the tables in the text are appropriate and easy to interpret while the references cited are relevant and most publications are recent.

To conclude, I recommend that the manuscript be accepted for publication, though the suggested changes to the results should be effected beforehand.

Author Response

Comments and Suggestions for Authors:

A well presented manuscript representing a systematic review on identifying and categorization of predictors of CRF based on a theoretical framework of 3P model i.e. predisposing factors, precipitating factors and perpetuating factors.

It is correct, that the results of this review could stimulate future research employing emerging technologies to provide more refined theoretical basis of assessment and screening of CRF and this in turn could lead to a more accurate and personalized treatment of clinical work.

Despite the limitations elaborated in the paper, particularly the need for additional studies to corroborate the value of defined CRF predictors, it is plausible, as suggested, that assessment and management of CRF should also focus on factors of the 3P model especially the controllable predisposing factors to improve the quality of life of cancer survivors.

The approach is scientifically sound and research methods are appropriate. However, though the results are well presented there are areas where interpretations and conclusions are included in the same results instead of being in the discussion and or conclusion. On the other hand, the figures and the tables in the text are appropriate and easy to interpret while the references cited are relevant and most publications are recent.

To conclude, I recommend that the manuscript be accepted for publication, though the suggested changes to the results should be effected beforehand.

Re: We feel great thanks for your professional review work on our article. As you are concerned, there are several problems that need to be addressed. According to your nice suggestions, we have made extensive corrections to our previous manuscript, especially the results and discussion. These changes will not influence the content and framework of the paper. And here we did not list the changes but highlighted or marked in red in revised manuscript.

Thank you again for your positive comments and valuable suggestions to improve the quality of our manuscript.

Reviewer 2 Report

Comments and Suggestions for Authors

Thank you for the opportunity to review this article. Overall, the manuscript is well designed and effectively addresses important topics related to fatigue in cancer patients. However, there are a few areas where editing can be beneficial.

Firstly, it is recommended to ensure that the reference writing in lines 41 and 42 of the introduction aligns with the framework of the journal. This will help maintain consistency and adherence to the journal's guidelines.

Additionally, in lines 117 and 131, consider using name abbreviations to enhance clarity and conciseness.

Furthermore, it would be helpful to provide an explanation for why pain is categorized into psychological characteristics, as mentioned in the text.

Considering these suggestions, I believe the article can be accepted and printed in its current form with the necessary revisions.

Author Response

Comments and Suggestions for Authors:

Thank you for the opportunity to review this article. Overall, the manuscript is well designed and effectively addresses important topics related to fatigue in cancer patients. However, there are a few areas where editing can be beneficial.

Firstly, it is recommended to ensure that the reference writing in lines 41 and 42 of the introduction aligns with the framework of the journal. This will help maintain consistency and adherence to the journal's guidelines.

Additionally, in lines 117 and 131, consider using name abbreviations to enhance clarity and conciseness.

Furthermore, it would be helpful to provide an explanation for why pain is categorized into psychological characteristics, as mentioned in the text.

Considering these suggestions, I believe the article can be accepted and printed in its current form with the necessary revisions.

Respond to Reviewer#2’ comments:

  1. Firstly, it is recommended to ensure that the reference writing in lines 41 and 42 of the introduction aligns with the framework of the journal. This will help maintain consistency and adherence to the journal's guidelines.

Re: Thank you for pointing out that we did not properly cite the literature as required by the journal. The reference formatting has been revised as per the journal's guidelines. The modified sentences can be found in the revised manuscript, as shown below:

“CRF is a frequently reported symptom, with a prevalence of 45 to 80% among all cancer patients [2,3], with a particular emphasis on those undergoing radiotherapy or chemo-therapy [4].”

  1. Additionally, in lines 117 and 131, consider using name abbreviations to enhance clarity and conciseness.

Re: Thank you for your kind reminder. We've changed the names to abbreviations to enhance clarity and conciseness:

“In this study, data that satisfied the aforementioned criteria were extracted and saved in a Microsoft Excel spreadsheet by two researchers (Y.W. and X.L.).”

“The assessment of study quality was independently conducted by two reviewers (Y.W. and H.Z.).”

  1. Furthermore, it would be helpful to provide an explanation for why pain is categorized into psychological characteristics, as mentioned in the text.

Re: We think this is an excellent suggestion and prompts us to rethink pain categorization. It has been argued that pain is not only a physical symptom but also a psychological feeling, so we categorized it as a psychological characteristic in the first submitted manuscript. However, considering two reviewers' suggestion, we think it is more appropriate to categorize pain, lack of appetite, feeling drowsy, dyspnea, and urinary dysfunction as physical symptoms. The following text has been added in the revised manuscript:

“3.2.5. Physical symptoms

The included literature identified pain, lack of appetite, feeling drowsy, dyspnea, and urinary dysfunction as physical symptom dimensions that could predict fatigue. Hwang et al. explored the sources of predictor variables of fatigue in different dimensions. They found that physical symptoms such as pain, lack of appetite, feeling drowsy, and dyspnea served as a unidimensional predictor of fatigue in patients with mixed cancers and functioned as the sole independent predictor in the multidimensional model [30]. The results of another three studies in patients with different types of cancer showed that pain is positively correlated with fatigue severity[29,37,40] ; however, pain cannot be used as an independent predictor of fatigue.”

We tried our best to improve the manuscript and made some changes in the manuscript. These changes will not influence the content and framework of the paper. And here we did not list the changes but highlighted or marked in red in revised manuscript.

We appreciate for Reviewers' warm work earnestly, and hope that the correction will meet with approval.

Once again, thank you very much for your comments and suggestions.

Reviewer 3 Report

Comments and Suggestions for Authors

The introduction identifies limitations of patient reported fatigue but given that CRF is subjective it is not clear how other approaches to assessment would be more appropriate.  There is also a statement that clinicians do not recognise the importance of fatigue but this is rather vague and needs clarification as to why it is important.  The following sentence also requires further explanation and supporting references: "However, most of their predictors come from single electronic health records (EHR), which is not comprehensive enough and lacks systematic selection criteria."  Further to this, it is not clear what is meant by "they were single and one-sided" in relation to the criticism of existing literature.  If indeed, the existing literature has such great limitations then surely primary rather than secondary research would be more appropriate to address the aim.  It is also not clear why studies of the "genomic domain" have been excluded as this potentially provides a partial picture of the predisposing factors.

Methods: If the full search terms are provided in an appendices this should be referenced in the methods.  Please clarify which researchers evaluated the studies as the following statement is not clear "Two researchers (Yiming Wang et al.) independently evaluated the 94 articles...".  The same point applies to the assessment of study quality.  There are no details provided of data analysis within the methods.

Results: 22,839 studies were reported to be found in the search but the subsequent sentence only accounts for 15,799.  The PRISMA flowchart indicates reasons for exclusion at full text stage but the reasons lack clarity as it is suggested that they were all 'irrelevant'.  The reasons provided need to relate back to the review inclusion/exclusion criteria and where multiple reasons are provided for a single study this requires explanation (to make clear why the overall numbers exceed the number of studies excluded).  

It is not clear from the methods or results how the categories have been developed.

Line 161 states: "Some studies have also defined CRF based on the questionnaire:" this statement needs to clarify that fatigue severity has been categorised in this way with cut off scores for minimal clinically important differences.  

Line 251-253: The following sentence does not make sense and CSF has not been previously defined:  "Another study, also in breast cancer patients, showed that increased upper limb volume (80% of total limb length measured from the ulnar styloid to the tip of the acromion) was associated with CSF as a predictor of it"

The categorisation of pain as a psychological trait is rather unusual and questionable.

page 261-262:  the delineation of different dimensions of fatigue has not been considered up to this point and is likely to be a key factor in the identification of the 3Ps.  Without consideration of the individual dimensions it is unlikely that a clear picture of the 3Ps can be established.

Section 3.3 - it is not clear why the opening paragraph to this section refers to predisposing factors.  Is this a typographical error?

Line 310-311: the statement "The most common predisposing factors of CRF are inflammation and immune response." is made on what basis?  Should this be that they are the most commonly studied?  If we already know that they are the most common predisposing factors this should be included in the background.

Page 351: what is CSF?  Should this be CFS and if this is the case the relationship between CRF and CFS needs to be explored.

Page 366-368:  The claim that: "Specifically, patients with higher levels of fatigue are less physically active, which may subsequently lead to body dysmorphic disorder, which further exacerbates the persistence of fatigue" requires supporting evidence.  Which of the preliminary studies has shown this?

Discussion

Much of the discussion is repetition of the results. It would be useful to consider the multidimensional nature of fatigue in more depth and the limitations of the evidence to date.  Further to this it would be useful to compare the findings for CRF to the 3Ps for fatigue in other long term conditions to explore whether there are key differences.  Whilst a range of factors have been considered within the primary data, identification of potential gaps needs to be considered. Rather than simply indicating that further research is needed it would be helpful to consider how the research could best be conducted to address the existing limitations of the evidence.

Line 432-433: The claim that "CRF is more common in breast cancer survivors than in other cancer survivors..." is not supported by the reference provided.  In addition, the statement that "Breast Cancer-Related Fatigue (BCRF) is a multidimensional concept affecting physical, cognitive, and emotional domains" may well be true but is not specific to breast cancer as is suggested.

There was reference to appendices that do not appear to have been included.

There are a number of errors in the presentation of the reference list.

Comments on the Quality of English Language

The entire paper needs to be carefully checked to ensure that meaning is accurate.  For example the statement "A total of 27 papers met the inclusion and exclusion criteria." suggests that the papers would not be included as they 'met' the exclusion criteria.

The following sentence implies that fatigue has malignancies: "Study participants were cancer patients with fatigue due to various malignancies and their treatment (including solid and hematologic malignancies)."

Other queries regarding English language are included in the scientific feedback.

Author Response

Respond to Reviewer#3’s comments:

  1. The introduction identifies limitations of patient reported fatigue but given that CRF is subjective it is not clear how other approaches to assessment would be more appropriate.

There is also a statement that clinicians do not recognize the importance of fatigue but this is rather vague and needs clarification as to why it is important.

Re: Thank you for pointing out that we did not describe this adequately. We modified the sentence related to the management and importance of CRF:

“The pathogenesis of CRF is unclear... The most popular and supported hypothesis includes the inflammatory hypothesis, which states that cancer and cancer treatments activate the immune system to release pro-inflammatory factors that affect the central nervous system, resulting in symptoms such as sleep disturbances, fever, and severe fatigue[5,7].

In the state of fatigue, the patient's immune function is reduced and susceptible to infections, as well as feeling weak and discouraged, seriously affecting their therapeutic effect and quality of life and even increasing the risk of death[4,12,13]. Effective early screening and fatigue assessment are essential for these patients [14]. Although CRF management is strongly recommended in guidelines and the literature, its implementation in clinical practice is often lacking, leading to underestimation and under-treatment [15–17]. The primary barrier to implementing CRF management is the lack of accurate knowledge of care providers about fatigue and its treatment options and the effects of fatigue in patients [18]. Furthermore, patients often do not voluntarily report this symptom for fear of interfering with treatment or because they feel that fatigue is unavoidable [14,19].”

  1. The following sentence also requires further explanation and supporting references: "However, most of their predictors come from single electronic health records (EHR), which is not comprehensive enough and lacks systematic selection criteria." Further to this, it is not clear what is meant by "they were single and one-sided" in relation to the criticism of existing literature. If indeed, the existing literature has such great limitations then surely primary rather than secondary research would be more appropriate to address the aim.

Re: Thanks for your comments. Given the limitations of the current primary study, we think that categorizing predictors of CRF based on the 3P model could provide some reference for further research. As suggested by the reviewer, corresponding references have been added to modify and explain this section.

“With the rapid development of artificial intelligence, several studies have shown that prediction models based on machine learning algorithms can be a good aid for early screening and assessment of diseases[20] and have promising applications in CRF[21,22]. The selection of predictors, which serve the accuracy and interpretability of the model to a certain extent, is the primary foundation of constructing a prediction model. The selection of easily accessible electronic health records (EHRs) for modeling[23] purposes is a common practice observed in the previous research. Nevertheless, predictive models may need to be improved in their applicability and accuracy due to variations in biology, genetics, and environmental factors among different populations[24]. Thus, a comprehensive and systematic criterion for selecting predictors is required in similar studies in the future.”

  1. It is also not clear why studies of the "genomic domain" have been excluded as this potentially provides a partial picture of the predisposing factors.

Re: Thanks for your suggestion. This section was explained in line 594 of the discussion. First, there was a recent review that has conducted research on fatigue in the genomic domain and drawn comprehensive conclusions. Furthermore, one of the aims of this review is to categorize the complexity of predictors according to the 3Ps, and studies in the genomic domain are more clearly predominantly predisposing factors. In summary, we did not include literature in the genomic domain.

“Second, we should have included literature studying the association of genetic polymorphisms with CRF because research involving these aspects already exists[26]

  1. Methods: If the full search terms are provided in an appendices this should be referenced in the methods. Please clarify which researchers evaluated the studies as the following statement is not clear "Two researchers (Yiming Wang et al.) independently evaluated the 94 articles...". The same point applies to the assessment of study quality. There are no details provided of data analysis within the methods.

Re: Thanks for your kind reminder. We will upload the complete search strategy according to your suggestion. Additional information on the development of classification criteria has been provided in the 2.3 Data extraction section of the original text. And we have re-written researchers’ names to enhance clarity and conciseness:

“All search strategies were determined after several pre-searches (see Supplementary Material for details of the search strategies.) The PRISMA flowchart showed the detailed strategies for paper search and screening. Y.W. and L.T. independently evaluated the articles regarding the inclusion and exclusion criteria (see below).”

“The predictors were classified into predisposing, precipitating, and perpetuating factors according to the 3P model defined by Sleight et al.[25]. The precipitating and perpetuating factors were determined solely based on the theory above. According to the research of Hwang et al.[30], the predisposing factors were secondary categorized into baseline fatigue, demographic characteristics, clinical characteristics, psychosocial traits, and physical symptoms.”

“The assessment of study quality was independently conducted by two reviewers (Y.W. and H.Z.)

  1. Results: 22,839 studies were reported to be found in the search but the subsequent sentence only accounts for 15,799. The PRISMA flowchart indicates reasons for exclusion at full text stage but the reasons lack clarity as it is suggested that they were all 'irrelevant'. The reasons provided need to relate back to the review inclusion/exclusion criteria and where multiple reasons are provided for a single study this requires explanation (to make clear why the overall numbers exceed the number of studies excluded).

Re: Thank you for pointing this out. This section was not well described, and the original text and Figure 1 have been revised accordingly.

“An initial search of electronic databases identified 22,839 articles for review. After removing 6719 duplicates, 16120 documents were retained. 8892 articles were excluded based on title, keywords, and abstract, and 188 full-text articles were reviewed for eligibility. Ultimately, 27 articles published between 2002 and 2023 were included in this systematic review based on our inclusion criteria.”

  1. It is not clear from the methods or results how the categories have been developed.

Re: Thanks for your comments. Additional information on the development of classification criteria has been provided in the 2.3 Data extraction section of the original text.

“The predictors were classified into predisposing, precipitating, and perpetuating factors according to the 3P model defined by Sleight et al.[25]. The precipitating and perpetuating factors were determined solely based on the theory above. According to the research of Hwang et al.[30], the predisposing factors were secondary categorized into baseline fatigue, demographic characteristics, clinical characteristics, psychosocial traits, and physical symptoms.”

  1. Line 161 states: "Some studies have also defined CRF based on the questionnaire:" this statement needs to clarify that fatigue severity has been categorised in this way with cut off scores for minimal clinically important differences.

Re: We think this is an excellent suggestion. We have re-written this part as suggested by the reviewers:

“Most of the studies analyzed fatigue scores as continuous variables. Based on this criterion, some of them transformed into categorical variables considering the minimal clinically important differences (MCID) as a cut-off score, differentiating between changes in fatigue as clinically significant or not, or using a boundary value to classify the severity of fatigue.”

  1. Line 251-253: The following sentence does not make sense and CSF has not been previously defined: "Another study, also in breast cancer patients, showed that increased upper limb volume (80% of total limb length measured from the ulnar styloid to the tip of the acromion) was associated with CSF as a predictor of it"

Re: Thanks for your valuable comments The definition of CSF has already been given in the previous section. This sentence aims to point out a predictor of CSF, i.e., increased upper limb volume, which is categorized as a clinical characteristic, as explained in the summary at the beginning of the paragraph.

“In different studies, the designation of clinically significant fatigue (CSF) was defined as BFI scores equal to or greater than 4[28], similar to VNR scores[41].”

“Clinical characteristics of patients, such as disease stage, tumor size, upper limb volume, and comorbidities, are also closely related to cancer-related fatigue.”

  1. The categorisation of pain as a psychological trait is rather unusual and questionable.

Re: We think this is an excellent suggestion and prompts us to rethink pain categorization. It has been argued that pain is not only a physical symptom but also a psychological feeling, so we categorized it as a psychological characteristic in the first submitted manuscript. However, considering two reviewers' suggestion, we think it is more appropriate to categorize pain, lack of appetite, feeling drowsy, dyspnea, and urinary dysfunction as physical symptoms. The following text has been added in the revised manuscript:

“3.2.5. Physical symptoms

The included literature identified pain, lack of appetite, feeling drowsy, dyspnea, and urinary dysfunction as physical symptom dimensions that could predict fatigue. Hwang et al. explored the sources of predictor variables of fatigue in different dimensions. They found that physical symptoms such as pain, lack of appetite, feeling drowsy, and dyspnea served as a unidimensional predictor of fatigue in patients with mixed cancers and functioned as the sole independent predictor in the multidimensional model [30]. The results of another three studies in patients with different types of cancer showed that pain is positively correlated with fatigue severity[29,37,40] ; however, pain cannot be used as an independent predictor of fatigue.”

  1. page 261-262: the delineation of different dimensions of fatigue has not been considered up to this point and is likely to be a key factor in the identification of the 3Ps. Without consideration of the individual dimensions it is unlikely that a clear picture of the 3Ps can be established.

Re: Thanks for your precious suggestions. We have considered the different dimensions of fatigue, which, as you say, could be a critical factor in identifying the 3P factors. However, the assessment tools used in the included studies varied, and some unidimensional scales existed, limiting the feasibility of the above idea. Therefore, this was not explored in the original article, and future research could focus on this direction.

  1. Section 3.3 - it is not clear why the opening paragraph to this section refers to predisposing factors. Is this a typographical error?

Re: We are really sorry for our careless mistakes. Thanks for your kind reminder.

“For precipitating factors, most of the included studies focused on exploring the underlying immune and inflammatory factors for CRF…”

  1. Line 310-311: the statement "The most common predisposing factors of CRF are inflammation and immune response." is made on what basis? Should this be that they are the most commonly studied? If we already know that they are the most common predisposing factors this should be included in the background.

Re: Thank you for your careful reading. We apologize for the typographical error and unclear statement in this section. We have rewritten the sentences.

“The CRF predictors categorized as precipitating factors were concentrated on inflammation and immune response.”

  1. Page 351: what is CSF? Should this be CFS and if this is the case the relationship between CRF and CFS needs to be explored.

Re: Thank you for your valuable comments. The word here is CSF, clinically significant fatigue, as defined in a previous article.

“In different studies, the designation of clinically significant fatigue (CSF) was defined as BFI scores equal to or greater than 4[28], similar to VNR scores[41].”

  1. Page 366-368: The claim that: "Specifically, patients with higher levels of fatigue are less physically active, which may subsequently lead to body dysmorphic disorder, which further exacerbates the persistence of fatigue" requires supporting evidence. Which of the preliminary studies has shown this?

Re: As suggested by the reviewer, we have added more references to support this idea.

“Specifically, patients with higher levels of fatigue are less physically active, which may subsequently lead to body dysmorphic disorder, exacerbating the persistence of fatigue[59].”

PMID: 18765328  DOI: 10.1188/08.ONF.815-821

  1. Much of the discussion is repetition of the results. It would be useful to consider the multidimensional nature of fatigue in more depth and the limitations of the evidence to date. Further to this it would be useful to compare the findings for CRF to the 3Ps for fatigue in other long term conditions to explore whether there are key differences. Whilst a range of factors have been considered within the primary data, identification of potential gaps needs to be considered. Rather than simply indicating that further research is needed it would be helpful to consider how the research could best be conducted to address the existing limitations of the evidence.

Re: Thank you for your comments on the discussion; we greatly benefited. This part has been revised according to the reviewers' suggestions, especially the section on limitations.

“Unidimensional questionnaires similar to the BFI have fewer questions, which are easier to use. However, the lack of predictors for different dimensions of fatigue has somewhat hindered the exploration of predictors for different dimensions of fatigue using the 3P model; out of the 27 studies included, only 11 defined fatigue, which needed to be more consistent. Most definitions differentiate the presence or absence of fatigue by a specific cut-off value of the questionnaire scores. Agarwal et al. and Rau-donis et al. differentiated the severity of fatigue[8,49]. In contrast, Feng et al. compared the value of change in fatigue using the minimum clinically important difference to ex-amine a clinically meaningful difference[37,39]. The lack of uniform definitional criteria to define fatigue may lead to non-comparability and confounding of results.”

“Furthermore, most of the included literature was found in cross-sectional and longitudinal studies without a control group, thus, we could not determine whether the associations between these predictors and fatigue were specific to disease diagnosis and/or treatment. Fatigue is a common symptom in the community, with up to half of the general population reporting fatigue in extensive surveys rather than being specific to cancer patients. Chronic fatigue syndrome (CFS) is a medical condition distinguished by severe and incapacitating fatigue that persists for a minimum of six months and is accompanied by various symptoms like rheumatologic, infectious, and neuropsychiatric[96]. While there may be similarities with the mechanisms underlying CRF, such as inflammation and permanent deconditioning[97], they are not precise. Hence, it could be advantageous to conduct a comparative analysis of CRF results and fatigue in other prolonged conditions using the 3P model as a framework to determine whether significant distinctions exist.”

  1. Line 432-433: The claim that "CRF is more common in breast cancer survivors than in other cancer survivors..." is not supported by the reference provided. In addition, the statement that "Breast Cancer-Related Fatigue (BCRF) is a multidimensional concept affecting physical, cognitive, and emotional domains" may well be true but is not specific to breast cancer as is suggested.

Re: Thank you for your suggestions. This part of the statement needs to be more accurate and has been revised.

“A recent review reaffirmed that CRF is the most prevalent and distressing symptom in breast cancer survivors after treatment[71]. Moreover, another study highlighted that Breast Cancer-Related Fatigue (BCRF) is a multidimensional concept that impacts physical, cognitive, and emotional domains[72].”

  1. There was reference to appendices that do not appear to have been included.

Re: Thank you for your careful reading, and if you are referring to supplemental material that includes a search strategy, we will submit it with the manuscript.

  1. There are a number of errors in the presentation of the reference list.

Re: Thanks for your kind reminder. We have carefully checked the reference list and made some changes.

  1. The entire paper needs to be carefully checked to ensure that meaning is accurate. For example, the statement "A total of 27 papers met the inclusion and exclusion criteria." suggests that the papers would not be included as they 'met' the exclusion criteria.

The following sentence implies that fatigue has malignancies: "Study participants were cancer patients with fatigue due to various malignancies and their treatment (including solid and hematologic malignancies)."

Other queries regarding English language are included in the scientific feedback.

Re: Thanks for your suggestion. We invited a friend of us who is a native English speaker from the USA to help polish our article. And here we did not list the changes but marked in red in the revised manuscript. We appreciate for Editors/Reviewers' warm work earnestly and hope that the correction will meet with approval.

We tried our best to improve the manuscript and made some changes in the manuscript. These changes will not influence the content and framework of the paper. And here we did not list the changes but highlighted or marked in red in revised manuscript.

We appreciate for Reviewers' warm work earnestly, and hope that the correction will meet with approval.

Once again, thank you very much for your comments and suggestions.
